# Insights into Clinical Disorders in Cowden Syndrome: A Comprehensive Review

**DOI:** 10.3390/medicina60050767

**Published:** 2024-05-06

**Authors:** Lorin-Manuel Pîrlog, Andrada-Adelaida Pătrășcanu, Mariela Sanda Militaru, Andreea Cătană

**Affiliations:** 1Department of Molecular Sciences, Faculty of Medicine, University of Medicine and Pharmacy “Iuliu Hațieganu”, 400012 Cluj-Napoca, Romania; lorin.pirlog@gmail.com (L.-M.P.); dr.mariela.militaru@gmail.com (M.S.M.); catanaandreea@gmail.com (A.C.); 2Regional Laboratory Cluj-Napoca, Department of Medical Genetics, Regina Maria Health Network, 400363 Cluj-Napoca, Romania; 3Department of Oncogenetics, “Prof. Dr. I. Chiricuță” Institute of Oncology, 400015 Cluj-Napoca, Romania

**Keywords:** Cowden syndrome, PTEN, clinical disorders, review

## Abstract

PTEN Hamartoma Tumour Syndrome (PHTS) encompasses diverse clinical phenotypes, including Cowden syndrome (CS), Bannayan–Riley–Ruvalcaba syndrome (BRRS), Proteus syndrome (PS), and Proteus-like syndrome. This autosomal dominant genetic predisposition with high penetrance arises from heterozygous germline variants in the PTEN tumour suppressor gene, leading to dysregulation of the PI3K/AKT/mTOR signalling pathway, which promotes the overgrowth of multiple and heterogenous tissue types. Clinical presentations of CS range from benign and malignant disorders, affecting nearly every system within the human body. CS is the most diagnosed syndrome among the PHTS group, notwithstanding its weak incidence (1:200,000), for which it is considered rare, and its precise incidence remains unknown among other important factors. The literature is notably inconsistent in reporting the frequencies and occurrences of these disorders, adding an element of bias and uncertainty when looking back at the available research. In this review, we aimed to highlight the significant disparities found in various studies concerning CS and to review the clinical manifestations encountered in CS patients. Furthermore, we intended to emphasize the great significance of early diagnosis as patients will benefit from a longer lifespan while being unceasingly advised and supported by a multidisciplinary team.

## 1. Introduction

### 1.1. Definition

PTEN Hamartoma Tumour Syndrome (PHTS) presents a range of clinical phenotypes, including Cowden syndrome (CS, OMIM 158350), Bannayan–Riley–Ruvalcaba syndrome (BRRS, OMIM 153480), Proteus syndrome (PS, OMIM 176920), and Proteus-like syndrome (PS-like) [1,2].

### 1.2. Pathophysiology of the PTEN

PHTS is an autosomal dominant genetic predisposition caused by heterozygous germline variants in the PTEN tumour suppressor gene, localised on 10q23.31 [3,4]. The PTEN gene is endowed with a double activity: lipid phosphatase and protein phosphatase at the cytoplasmic level. Concomitantly, PTEN has a paramount role in genomic stability when acting at the nuclear level by maintaining the chromosomal architecture as well as supervising the cell cycle. Regarding its action at the cytoplasmic level, as a lipid phosphatase, PTEN plays a negative feedback role in the PIP3/AKT/mTOR signalling pathway [5]. Typically, the cascade is initiated by the binding of signal molecules to growth factor tyrosine kinase (RTK) receptors. When activated, the growth factors activate phosphatidylinositol 3-kinase, which phosphorylates phosphatidylinositol 4,5-diphosphate (PIP2) to the second messenger phosphatidylinositol 3,4,5-trisphosphate (PIP3), switching on the serine/threonine protein kinase B (AKT) [6]. Successively, AKT suppresses the tuberous sclerosis complex (TSC), inhibiting the mTOR (mechanistic target of rapamycin) complex. Through its lipid phosphatase function, PTEN dephosphorylates the secondary messenger molecule PIP3 to PIP2, effectively impeding AKT activation. Therefore, the mTOR complex eagerly activates, causing abnormal cell proliferation and transformation [7].

Regarding its protein phosphatase activity, PTEN also possesses a negative feedback role in the RAS/MAPK pathway by inhibiting Grb2 (growth factor receptor bound protein 2), which connects growth factor receptors and the RAS/MAPK signalling pathway by recognizing the SH (Src homology) domains: one SH2 domain and two SH3 domains [7,8]. Through the SH2 domain, which may be recognized as the prototypical modular protein–protein interaction domain, it attaches to the activated RTK receptor [9]. Moreover, through SH3, a small protein domain, defined as a conserved sequence in the viral adaptor protein v-Crk [10], PTEN interacts with one end of SOS (son-of-sevenless), a dual specificity guanine nucleotide exchange factor (GEF) that regulates the Ras and Rho family, promoting the integrity of DNA; the other end of the SOS protein connects with a domain called Ras-GEF, which leads to RAS and RAF molecule activation and downstream phosphorylation of MERK 1 and 2, thus activating ERK1 and ERK2 with a significant function in cell growth and cell proliferation [9,10,11].

### 1.3. Pathophysiology of the PHTS

PTEN plays a crucial role in suppressing the PI3K/AKT/mTOR signalling cascade, regulating cell growth. Dysfunctional PTEN leads to overgrowth through pathway dysregulation. It leads to multi-system tissue and cell overgrowth with highly diverse phenotypic presentations, encompassing both benign and malignant tumours, macrocephaly, developmental delay, autism spectrum disorders, thyroid abnormalities, and skin lesions [4,11]. The syndrome is associated with elevated risks of various cancers, with reported increased lifetime risks and earlier onset for breast, endometrial, thyroid, and other cancers [1,2]. Hereditary cancers carry unique significance as they tend to develop earlier and confer a heightened risk of aggressive, multifocal, and bilateral cancers [12]. About 5–10% of malignancies result from hereditary predisposition syndromes, highlighting their substantial impact [13,14]. These syndromes, often stemming from mutations in key genes involved in cell regulation and DNA repair, elevate cancer risk, as exemplified by CS and PTEN mutations [15]. The median age of cancer diagnosis in patients diagnosed with CS is around 36 years, with cumulative lifetime risks of 85–90% in females and 81–88% in males [1]. Also, the mean age of patients diagnosed with CS varies in the literature, being reported between 36 and 50.5 years [11,16,17,18,19].

Early diagnosis is critical for inclusion in PHTS-specific surveillance programs, aiding in timely cancer detection, improved prognosis, and extended life expectancy [1,2,11]. However, PHTS remains under-recognized due to its complex and variable clinical presentation, and its low prevalence of approximately 1 in 200,000 is believed to be underestimated [20,21].

The rate of de novo pathogenic PTEN variants exhibits variability across different literature sources, with certain authors citing rates ranging from 10% to 40% [22], while others report figures of 10% to 44% [2,23]. Moreover, there are indications from some sources of a broader range spanning from 10% to 47% [1].

Research findings indicate that a substantial portion, approximately 80% [11] and potentially even as high as 85% [2,12], of individuals fitting the stringent criteria for CS exhibit germline PTEN mutations. Furthermore, closer examination reveals that around 30–35% of patients fulfilling the diagnostic criteria carry these mutations [24]. Additionally, investigations suggest that the proportion of patients who possess these mutations while adhering to the less stringent diagnostic criteria could be as much as 25% [2,12,23]. Germline PTEN mutations are also associated with BRRS and PS [2,25].

After the identification of the PTEN gene in 1997 [26], it became clear that not every patient with PHTS possesses pathogenic PTEN mutations in their germline. Roughly 15% of individuals with classic CS and about 95% of those with CS-like (not fulfilling complete diagnostic criteria) presentations do not exhibit detectable PTEN mutations [12]. Patients diagnosed with CS/CS-like presentations who exhibit nonmutant PTEN alleles may carry mutations in alternative genes, including SDHx, KLLN, AKT1, PIK3CA, PIK3R1/2, WWP1, SEC23B, and USF3 [12,13,23,27].

In the subsequent sections, we offer a comprehensive overview of both malignant and benign manifestations of CS, as well as delve into the genotype–phenotype relationship.

## 2. Clinical Disorders

CS gives rise to a multitude of disorders affecting nearly all body systems. Given its intricate nature, the upcoming sections provide insight into and outline the array of clinical issues experienced by individuals diagnosed with CS.

### 2.1. Breast Disorders

Individuals affected by CS face an increased susceptibility to both benign and malignant breast tumours [23]. While breast cancer (BC) was initially not considered a typical part of the syndrome, it has since emerged as the predominant malignancy within the CS spectrum [24].

The onset of BC in CS patients typically occurs at an age range spanning from 38 and 46 years [15,23,28], though some reports indicate a broader span of 38 to 50 years [24], and others even extend to 52 years [20]. Notably, certain authors suggest an even earlier onset around the age of 30 [2]. In comparison, the average age of BC diagnosis in the general population is 63 years [20].

CS women are reported to exhibit a 25% to 50% risk of developing BC [11,23], which is a notable contrast to the 12% risk observed among women in the general population [11]. Recent studies, however, have revealed a significantly greater risk of approximately 77–85% [24,28,29]. The prevalence of bilateral BC in CS females ranges from 25% to 48%, markedly surpassing the 0.8–3% incidence in the general population. Moreover, CS patients are at a remarkable 9-fold higher risk of developing BC as a second primary malignancy [20], with a substantial 29% risk of a secondary BC within a decade of the initial diagnosis [29]. Females affected by CS face a potential risk of up to 67% for the development of benign breast conditions [2], such as fibrocystic breast disease [23]. This vulnerability is not exclusive to female CS patients, as male individuals with PTEN mutations have also been reported to develop BC [23]. However, while several instances of male BC with PTEN mutations have been documented [30,31,32], its association with CS remains unproven, with the largest reported cohorts not observing such an association [24].

It is essential to acknowledge the potential selection bias inherent in the cited studies due to cohort construction methods. Therefore, despite the unequivocal elevation of BC risk in individuals with PTEN mutations, the precise degree of this risk remains subject to debate, with a need for future studies designed to mitigate bias.

### 2.2. Gastrointestinal Disorders

Although the initial perspective suggested that PTEN mutations did not confer an elevated risk of colon cancer (CC), recent data have challenged this notion [24]. The prevalence of CC occurrences among CS patients has been reported to range from approximately 9% to 17% [2,24,27,29], while its manifestation is often marked by an early onset [24]. In contrast, the general population exhibits a much lower prevalence of CC, standing at only 5% [2]. 

The median age at which CC is diagnosed in CS patients exhibits variance across the literature, with some studies citing medians of 46 to 58 years [20], while others propose a lower median of 44 years [29]. Nonetheless, the youngest documented case of a CS patient with CC was only 28 years old [29]; this case is a notable contrast to the mean age of CC diagnosis in the general population, which stands at 73 years [20]. This disparity underscores the heightened CC risk associated with PTEN mutations, revealing a 2- to 3-fold increased risk in comparison to the general population [29]. Furthermore, CS patients face a potential up to 6-fold increased risk of secondary CC development [20].

Notably, individuals diagnosed with CC often exhibit pre-existing or concomitant colonic polyposis [2,23,24]. Intriguingly, a significant proportion of individuals who present at least one mutant PTEN allele, ranging from 90% to 95%, display colorectal polyps upon undergoing colonoscopy [2,23,24,33]. These polyps encompass diverse histologies, including hamartomas (present in 65.8% of cases), juvenile polyps, ganglioneuromas (hamartomatous tumours that originate in enteric nervous system cells [34]), adenomatous polyps, inflammatory polyps, leiomyomas, lipomas, lymphoid polyps, and hyperplastic polyps, even though the last ones are common in the general population [2,23,24,29,35]. Synchronous histologies can be observed within the gastrointestinal polyps of individuals with CS [24]. While polyps can be identified in various gastrointestinal locations, studies suggest that those in the colon notably escalate the risk of malignancy in CS patients [29]. Although three cases of gastric cancers in CS patients have been documented, they occurred in individuals over 60 years old, and the connection between these two pathologies remains uncertain [29].

In adult CS patients, the presence of oesophageal glycogenic acanthosis has been reported in 20% of cases, forming part of the diagnostic criteria [23,24,36,37].

### 2.3. Endometrial Disorders

Endometrial cancer (EC) is acknowledged as a significant component of CS, but its prevalence and clinical characteristics remain poorly defined. In the general population, the risk of developing EC ranges from 2% [29] to 2.6% [2,11,23]. However, among CS patients, the prevalence varies considerably, with reported risks spanning from 5–10% [11], 19–28% [13], to 21–28% [2,15,23]. Consequently, the overall risk of EC in CS patients is approximately 40 times higher than in the general population [20]. In individuals carrying PTEN mutations, even without a formal CS diagnosis, EC was detected in 7.6% to 17% of cases [24]. 

The relationship between age and EC risk presents conflicting findings. Some studies suggest a 1% risk at age 20, rising to 2% at age 30 [20]. Conversely, others indicate a 30% risk at age 60 [2,23] and a 19–28% risk at age 70 [13,24,29]. Notably, a statistically significant increase in risk is observed around age 25 [2,23], persisting beyond age 50 when compared to the general population [11]. This is supported by the median age of EC diagnosis in CS patients, which is 48 years, an age significantly younger than the general population’s median age of 68 years [13,20,29]. There are also case studies in the literature reporting the discovery of EC in adolescent CS patients [38]. 

Screening for EC in CS patients is a subject of controversy, as it has not demonstrated a reduction in mortality [13]. 

Additionally, individuals who present at least one mutant PTEN allele may experience benign gynaecological conditions, such as uterine fibroids [2,23].

### 2.4. Dermatological Disorders

Dermatological manifestations are a distinctive feature of CS. These skin and mucosal characteristics play a crucial role as clinical indicators for diagnosing CS. However, the prevalence and specific attributes of these dermatological signs have evolved and may exhibit variations among individual patients. The array of dermatological manifestations observed in individuals with CS encompasses a wide spectrum, including multiple mucocutaneous lesions, melanoma, papilloma, acral keratosis, lipomas, trichilemmomas, penile freckling, sclerotic fibromas, café-au-lait spots, vitiligo, acrochordons, and even acanthosis nigricans [4,23,24,29,39,40,41].

While earlier studies suggested that all CS patients displayed dermatological manifestations, contemporary perspectives within the scientific community lean towards viewing this as a potential overestimation. Furthermore, these studies indicated that dermatological manifestations were prevalent among paediatric CS patients, yet they often lacked specific age-related details [24]. According to existing literature, approximately 99% of CS patients exhibit dermatological lesions by the age of 30 [21]. However, the accuracy of this percentage remains uncertain due to discrepancies in the literature, with prevalence rates varying between 90% and 100% before the age of 30 [16,39]. Nonetheless, what remains evident is that most CS patients do develop dermatological disorders by the conclusion of their third decade of life.

It is noteworthy that certain authors have reported a skewed distribution among CS patients in middle age or older with orofacial manifestations, noting that a ratio of 2.3 to 1 favours women over men in this subgroup. However, older studies did not explicitly highlight any discernible disparities between the sexes in terms of dermatological features in CS [16].

Papillomatosis stands out as a distinctive feature of CS [16]. These papillomatous growths can manifest in various oral and pharyngeal locations, including the buccal, lingual, gingival, and labial mucosa, as well as the pharynx and larynx [16,24,36]. Remarkably, oral papilloma associated with CS tend to remain asymptomatic, and their dimensions are typically limited to a maximum diameter of 3 mm [21,24]. Notably, papillomatosis can even impart a scrotal appearance to the tongue [21,36]. However, it is important to note that the reported prevalence of oral papilloma exhibits a wide range, varying from 15.2% to 85% across different studies [21]. Given this substantial variation, it is imperative to exercise caution in interpreting these statistics, as biases may be present, necessitating further research to establish more accurate figures. Additionally, CS patients may also present with oral fibromas, albeit with a prevalence that ranges from 14% to 76%, demonstrating a similarly broad range of reported prevalence rates [24].

Acral keratosis, although less frequently documented, is another dermatological characteristic associated with CS [16]. These keratotic lesions exhibit a maximum diameter of 4 mm and primarily appear in palmoplantar regions, affecting both paediatric and adult CS patients, presenting with a verrucous appearance [24,36]. Importantly, acral keratosis may also occur in other areas of the body, such as the face and trunk [21,24,36]. While initial reports suggested a prevalence of 63–73% for acral keratosis, recent studies have indicated a broader prevalence range of 10.2% to 82%, reinforcing the need for caution when interpreting prevalence data due to the variability in reported figures [21].

Lipomas serve as a significant clinical criterion for diagnosing CS [24,41]. In the general population, these fatty tumours have an estimated prevalence of approximately 1%. However, when considering CS patients, recent studies have revealed a substantially higher prevalence, ranging from 34.6% to 56.7% [21]. Notably, the first reported instance of testicular lipomas in a CS patient was documented as recently as 2003, emphasizing the rarity of this type of lipoma in the absence of testicular neoplasms [24]. Furthermore, subcutaneous lipomas have also been described in the context of CS, although specific prevalence data for this manifestation remain unreported [41].

Mucosal neuromas, which represent hamartomas of the peripheral nerve sheath, have been documented in CS [21,24]. These neuromas can manifest in various locations, including the mouth, face, extremities, and trunk [21,24,36]. A notable characteristic of mucosal neuromas is their tendency to cause discomfort, particularly impacting oral hygiene [36]. While earlier reports suggested a prevalence range of 5–10.9% for these neuromas, more recent studies have reported a prevalence of 0%, highlighting the need for further investigation and potential variations in prevalence among different patient populations [21].

Penile pigmentation, characterized by pigmented speckled macules on the genitalia, is another dermatological manifestation frequently encountered in CS patients [41]. In the general population, the prevalence of such pigmentation is reported to be up to 15% [24]. However, in CS patients, the prevalence is significantly elevated, estimated to be approximately three times higher than that of the general population, with reported prevalence rates ranging from 48% to 53% [24]. Other studies have also corroborated these findings, reporting similar prevalence figures of 41% [21] or 48% [40]. Intriguingly, penile freckling can manifest at a very young age, even before one year of age [36].

The presence of multiple trichilemmomas is strongly indicative of a PTEN mutation [24]. These lesions can appear on various parts of the body, including the face, neck, axillae, hands, abdomen, and other regions [24,36,41]. Notably, trichilemmomas can be observed in patients under 18 years of age, with prevalence rates ranging from 6% to 25% in some studies [24] or even higher, reaching up to 38% in adults [21]. The clinical similarity of these lesions to other dermatological manifestations necessitates histopathological examination for accurate diagnosis [24].

Although generally rare, the literature includes case reports describing sclerotic fibromas of the skin in CS patients [24].

Melanoma has been observed in individuals diagnosed with CS, although the association and prevalence remain subject to limited available data in the literature, warranting further investigation for a more comprehensive understanding. 

In the general population, melanoma prevalence ranges from 2% [23,42] to 2.8% [23]. Among CS patients, the reported prevalence of melanoma varies, with estimates of around 6% [2,4,23,39]. Nevertheless, some studies suggest different figures, such as 1% [23], 2–6% [42], 5% [15], or even as high as 28.3% [29]. It is important to note that the latter prevalence figure comes with a wide 95% confidence interval of 7.6% to 35.4% [29], suggesting potential bias in the cohort data. Age appears to influence melanoma prevalence among CS patients. At the age of 20, the prevalence is approximately 0.4%, but this risk increases significantly, multiplying by 15 times by the age of 70 [20].

The median age for diagnosing melanoma in CS patients is approximately 40 years, which is notably younger than the general population, where the median age is 63 years [20]. However, there have been reports of melanoma in very young CS patients, with one case documented in a 3-year-old [2,4,23,43,44,45].

The risk of developing melanoma as a second cancer in CS patients appears to be seven times higher than in the general population, emphasizing the importance of yearly dermatological evaluations as recommended by specialists [44,45].

### 2.5. Thyroid Disorders

CS patients exhibit various thyroid manifestations, encompassing both benign and malignant disorders [46]. Among the benign thyroid disorders observed in CS patients are multinodular follicular adenomas, nodular hyperplasia, adenomatous nodules, Hashimoto thyroiditis, lymphocytic thyroiditis, C cell hyperplasia, and goitre [2,4,23,24,27,36,46]. Some studies suggest that approximately three-quarters of adult CS patients exhibit these manifestations [2,23], while others report a lower prevalence ranging from 30% to 68% [24]. In paediatric patients with CS, the prevalence of these benign thyroid manifestations is even lower, ranging from 2% to 14% [24]. Hashimoto thyroiditis, for instance, is found in 3% to 21% of individuals with PTEN mutations, in contrast to a 2% prevalence in the general population [24]. Thyroid nodules are a common occurrence in CS patients, with reported rates of up to 65%, and multinodular goitres have been found in up to 4% of cases [24]. Remarkably, the youngest CS patient diagnosed with benign thyroid manifestations was only 5 years old [36].

As previously mentioned, CS presents malignant manifestations, particularly in the form of epithelial thyroid neoplasia rather than medullary thyroid cancer [2]. Intriguingly, the medullary thyroid cancer type, which is found in the general population and multiple endocrine neoplasia syndrome, specifically type 2, has not been identified in CS patients to date [2,4].

Epithelial thyroid neoplasia, particularly the follicular and papillary types, is considered characteristic of CS [2,4,29]. Notably, the papillary form of epithelial thyroid neoplasia is twice as common as the follicular form [23,24]. CS patients face a lifetime risk of developing thyroid cancer, estimated at approximately 35% [2,4,15,23,27], or potentially even slightly higher at around 38% [11,20,24,36]. Some authors provide risk estimates with large 95% confidence intervals, ranging from 6% to 38% [2,42], or even 14% to 35% [4,29]. Despite the variations in risk estimates, it is evident that CS patients face a significantly increased lifetime risk of developing thyroid cancer, considering the general population’s estimated risk is only 1% [2,42,47].

Individuals with CS tend to be diagnosed with thyroid neoplasia at a much younger age, with a median age ranging from 31 to 37 years [4,11,20], which is significantly earlier than the general population, where the median age is 53 years [20]. Moreover, the risk of developing thyroid cancer in CS as a second cancer is six times higher than in non-CS individuals [20]. Some of the youngest CS patients diagnosed with thyroid neoplasia were just 6 and 7 years old [4,36], underscoring the importance of considering CS in paediatric patients, especially in cases involving males and a history of benign manifestations [2,23,36].

The presence of benign thyroid disorders in CS significantly increases the risk of developing thyroid neoplasia. In adult CS patients, the presence of these benign disorders contributes to a higher risk compared to the general population and accounts for 55.6% of thyroid neoplasia cases in CS patients [16].

Notably, CS can result from mutations in various genes. Interestingly, patients with mutations in the SDHx gene exhibit a higher prevalence of thyroid cancer than those with PTEN mutations, although the prevalence is lower when patients have mutations in both SDHx and PTEN genes [42].

### 2.6. Renal Disorders

Kidney complications have a recognized association with CS [3,47]. The literature reports both benign manifestations, such as renal cysts [22], and malignant ones, notably renal cell carcinoma (RCC) [15,29]. RCC is considered a minor diagnostic criterion for CS [3,29], and in the general population, the risk of developing RCC is approximately 1.6% [2,23,42]. However, in CS patients, the lifetime risk is significantly elevated, with estimates ranging from 34% to 35% [4,24,29]. A recent study reported a risk of 33.6% [27]. Nevertheless, some articles present a wide 95% confidence interval of 2% to 34% [2,20,23,42], underscoring the need for further research. In this context, individuals with CS have an elevated risk, approximately 30 times higher than non-CS individuals [20].

The median age for RCC diagnosis in CS patients is reported to be around 49 to 55 years [20]. However, some studies suggest a median age of 39 years [29], signifying a noteworthy 10-year difference. This discrepancy necessitates additional investigations. Nevertheless, RCC tends to manifest earlier in CS patients compared to the general population, where the median age for RCC diagnosis is 68 years [20]. The risk of developing RCC begins to increase notably around the age of 40 [2], but cases of RCC have been documented at earlier ages.

Regarding the histology of RCC in CS patients, the literature identifies two predominant types [2,29]. The papillary type is more prevalent than the chromophobe type [2,4,24]. Notably, ultrasonography is not particularly sensitive in detecting RCC, especially when the mass is small, with CT or MRI preferred in these situations [2,23].

### 2.7. Nervous System Disorders

Cerebral abnormalities have been established as potential manifestations in individuals diagnosed with CS [42,48]. These abnormalities encompass Lhermitte–Duclos disease (LDD) and meningiomas, both of which have been documented in the literature as clinical features associated with this syndrome, although they may also manifest in non-syndromic cases [2,24,27,49]. Notably, gliomas have not been reported in CS patients [42].

LDD is a rare, benign condition characterized by the gradual hamartomatous overgrowth of the cerebellum, involving the expansion of neurons into the granular and molecular layers [2,11]. In adult individuals, the presence of LDD can raise suspicion of CS and is considered a major diagnostic criterion [2,11,24]. However, the association between paediatric CS patients and LDD is not as robust as it is in adults [24]. Radiologically, this condition exhibits a distinct appearance often described as “tiger-striped” or “tigroid” [2,11]. The documented prevalence of LDD varies in the literature, with reports ranging from 1.8% to 6% [24]. Additionally, there are inconsistencies in the age of LDD diagnosis, with some studies suggesting onset between ages 20–39 or 30–49 [5,24,50]. However, cases of LDD have been reported in CS patients ranging from 4 to 75 years old [51,52]. Gender does not appear to influence the presence or absence of LDD, although these tumours tend to develop more frequently on the right side of the cerebellum [50]. Typically, LDD affects only one part of the cerebellum, but instances involving the vermis have also been described [11,27]. Patients with LDD may exhibit symptoms such as headaches, visual disturbances, nausea, increased intracranial pressure, ataxia, or cranial nerve palsies, with symptom manifestation dependent on the size of the cerebellar tumour mass [11,49,50,52]. These masses can remain stable over time or exhibit unpredictable rapid growth, and the sole treatment option is complete surgical removal to prevent potential recurrence [11].

Meningiomas have been described in individuals with CS, as mentioned earlier. However, given their relatively common occurrence in the general population, it remains unclear whether meningiomas are characteristic manifestations of CS [24,27]. The prevalence of meningiomas in CS individuals is approximately 8.25%, and there seems to be a gender-related aspect, with these tumours more commonly identified in male CS patients. In contrast, meningiomas are generally more prevalent in females in the non-CS population [27].

### 2.8. Vascular Disorders

CS patients exhibit numerous multifocal vascular disorders, specifically hamartomatous vascular malformations, as reported in the literature [24,27,53]. These manifestations encompass arteriovenous fistulas and haemangiomas, including cavernous haemangiomas, and can be observed in both paediatric and adult patients. They are considered minor diagnostic criteria for CS [4,11,24,54,55]. These vascular disorders have the potential to affect various organs, including the brains dura, skin, muscles, and adrenal glands [4,21,49,56,57]. Remarkably, one-third of mutant-PTEN patients present with vascular malformations [24]. Several studies report comparable findings in CS patients, with prevalence rates for vascular disorders ranging from 18% to 34% [11]. These figures are two to three times higher than the prevalence of such malformations in the general population, where the range is typically 5% to 10% [11]. Nevertheless, one study suggested that the prevalence of these malformations in CS patients is nearly equivalent to that in the general population, with rates ranging from 6.4% to 11.4% [21].

These malformations are believed to arise due to the influence of proteins encoded by PTEN, which regulate the function of vascular endothelial growth factor [11,27,58,59,60].

### 2.9. Neurodevelopmental and Autism Spectrum Disorders

In CS patients, neurodevelopmental disorders such as macrocephaly, developmental delay, and autism spectrum disorder (ASD) are prevalent manifestations from early childhood [11,23,40,61,62]. Studies indicate that individuals with ASD and macrocephaly have a notably increased likelihood of carrying a mutant PTEN gene, with 1 to 2 out of every 10 such cases linked to PTEN mutations [63,64]. While macrocephaly is recognized as a major diagnostic criterion, ASD is considered a minor criterion [11,41]. Neurodevelopmental features, including ASD, developmental delay, intellectual disability, and epilepsy, are observed in nearly all CS patients, with some presenting all these characteristics [23,29]. However, it is worth noting that certain studies report that the exact prevalence of these features remains undetermined [36].

Macrocephaly is highly prevalent, affecting 80% to 100% of CS patients [23,27]. This condition results from generalized megalencephaly, with the size of both neuron and glial cells contributing to this enlargement [65,66]. Some authors have noted the presence of nonspecific white matter abnormalities in the brains of CS patients, characterized by enlarged perivascular spaces. [36] Additionally, it has been observed that the severity of macrocephaly in ASD patients with mutant PTEN is greater than in ASD patients with wild-type PTEN [23].

### 2.10. Immune Disorders

Immune disorders are reported in mutant PTEN patients [22]. However, these conditions are a distinct feature of CS patients, affecting anywhere from one-quarter to half of them [22,67]. These immune disorders encompass a range of manifestations, including lymphopenia, aberrant B and T cell homeostasis and function, hypogammaglobulinemia, and dysgammaglobulinemia [2,22,36]. These immune irregularities can result in various clinical presentations, such as colitis, thymic lymphoid hyperplasia leading to airway obstruction and gastric tube-related anaemia, vasculitis, Hashimoto thyroiditis, haemolytic anaemia, pulmonary cysts, and eosinophilic esophagitis [36,68,69,70,71,72]. Elevated blood levels of lactate and other inflammatory markers have also been observed in CS patients [22].

### 2.11. Metabolic Disorders

CS patients commonly exhibit a range of metabolic disorders, spanning from obesity to insulin dysregulation [73]. These individuals often display heightened insulin sensitivity, which contributes to the development of obesity and an elevated body mass index (BMI) [2,23,36]. Paradoxically, this heightened insulin sensitivity results in a substantially reduced risk of developing type 2 diabetes, while increasing the susceptibility to obesity and cancer [2,36]. Researchers have linked this phenomenon to the loss of inhibition on the PI3K-AKT signalling pathway [23].

## 3. Discussions

### 3.1. Overview of the CS Clinical Disorders

CS highlights a broad spectrum of clinical disorders as outlined previously. Table 1 provides a reference list of the aforementioned manifestations. For deeper insights into their development risks, prevalences, and other factors, we advise referring to the preceding text; the table serves merely as a concise overview of the manifestations documented in existing CS literature. 

It is crucial to note that not all manifestations listed above serve as diagnostic criteria for CS. Therefore, please consult the subsequent section, where the accepted major and minor criteria for diagnosing PHTS/CS are described.

### 3.2. Diagnostic criteria for PTHS/CS

Over time, the diagnostic standards for CS have changed. To make the diagnosis of CS easier, the International Cowden Consortium Criteria were created in 1996. To help doctors identify patients with CS, these criteria listed four major and eight minor criteria (Table 2) [11]. A diagnosis of CS is verified using the 1996 criteria if an individual meets one of the following conditions: adult LDD; a particular number of mucocutaneous features; macrocephaly combined with another major criterion; one major criterion plus three minor criteria; or the presence of four minor criteria. Additionally, patients must meet the requirement of two key criteria to be diagnosed with CS, one of which must be macrocephaly or Lhermitte–Duclos disease [11,24].

The initial criteria were revised in 2013 because of several changes that researchers put into place, adding new components to both the major and minor criteria. Novel criteria were presented, including macular pigmentation of the glans penis and epithelial endometrial cancer as major criterion, and vascular anomalies or malformations, autistic disorder, colon cancer, oesophageal glycogenic acanthoses, testicular lipomatosis, and oesophageal glycogenic acanthoses as minor criteria. Additionally, certain criteria were elevated from the minor criteria to the major criteria category, such as mucocutaneous lesions and gastrointestinal hamartomas (Table 2 and Table 3).

The National Comprehensive Cancer Network (NCCN) has embraced these updated criteria, incorporating the revised PTEN hamartoma syndrome clinical criteria as the foundation for its version 3.2024 CS/PTEN hamartoma syndrome management guidelines (Table 3).

Individuals or families whose one member satisfies the updated criteria or has a PTEN mutation can be diagnosed using these new criteria. For a person to be diagnosed with CS in this situation, they must meet at least three major criteria, with one of them being one of the following: macrocephaly, gastrointestinal hamartomas, or Lhermitte–Duclos illness OR present two major and three minor criteria (Table 3). It is necessary to have the presence of any two major criteria, with or without minor criteria, OR one major and two minor criteria, OR three minor criteria (Table 3) for a diagnosis in families where one member meets the revised PHTS clinical diagnostic criteria or has a PTEN mutation [24].

### 3.3. Surveillance Cancer Guidelines for Patients with PHTS/CS

As mentioned earlier, individuals diagnosed with CS face a heightened risk of developing cancers at a younger age than the general population. Consequently, diligent surveillance of these patients is crucial for early detection and treatment of malignancies. To address this need, the European Reference Network on Genetic Tumour Risk Syndromes (ERN GENTURIS) and NCCN have developed cancer surveillance guidelines specifically tailored for CS patients (Table 4 and Table 5) [74,75,76].

Both sets of cancer surveillance guidelines focus on the same six cancers: breast, thyroid, renal, colorectal, melanoma, and endometrial. Setting aside this commonality, the two guidelines exhibit variations in their recommendations regarding the approach to supervision.

Therefore, concerning breast cancer, NCCN suggests commencing screening at 18 years of age with self-examination, followed by clinical evaluations starting at 25, or earlier if familial history warrants. It is notable that NCCN advises initiating mammography and MRI by age 30, a decade earlier than ERN GENTURIS recommends for mammography. Additionally, the variance in timing for these diagnostic examinations to be performed merits acknowledgement. 

Another significant contrast in the initiation age for surveillance is observed in thyroid monitoring. NCCN advises beginning surveillance at age 7, whereas ERN GENTURIS advocates for commencement from age 18.

Regarding melanoma, colorectal, and renal cancers, both consortia offer largely similar recommendations. Minor disparities exist in the commencement age for colon cancer follow-up, with NCCN emphasizing the significance of familial history once more. It is noteworthy that for melanoma, ERN GENTURIS suggests initiating dermatology consultations at age 30, whereas NCCN does not specify a particular age, leaving the timing to the discretion of individual cases.

Discrepancies emerge when comparing the two guidelines, particularly regarding endometrial cancer surveillance. ERN GENTURIS does not advocate for specific surveillance measures, delegating the monitoring of this cancer to physicians during clinical assessments. Conversely, NCCN recommends educating patients to recognize signs of endometrial cancer, prompting them to seek medical attention, followed by a biopsy if necessary. Additionally, NCCN suggests conducting annual or biennial biopsies starting at age 35, emphasizing the importance of proactive screening rather than relying solely on patient recognition of warning signs.

While the visions of the two consortia may not align perfectly, it is essential to recognize that both emphasize the importance of proactive surveillance for CS patients. Despite any differences, the overarching goal is to raise awareness and address serious pathologies associated with this condition, with the potential for significant positive outcomes.

While ERN GENTURIS and NCCN guidelines primarily address monitoring the mentioned pathologies, physicians should also prioritize surveillance of other conditions linked to CS (Table 1). The decision regarding their supervision plan should be left to the discretion of physicians overseeing CS patients, who should evaluate risks and benefits on a case-by-case basis.

### 3.4. Screening and Psychological Implications of CS

The goal of CS cancer surveillance is to identify cancer early to provide better curative treatments. A multidisciplinary approach and patient commitment are required by recommendations [74]. The possibility of developing numerous cancers emphasises the necessity of ongoing surveillance, even when the illness carries a low financial cost to health [74]. For those who are 50% at risk, genetic testing is advocated, and for those who have been diagnosed, personalised surveillance is advised based on personal and family history [36]. But baseline risk evidence quality is judged inadequate, emphasising the necessity of worldwide and national registries for prospective data collecting [74]. Understanding cancer risk factors, individualised risk assessments, prophylactic medications, and customised treatments should be the main goals of research. Information groups play a crucial role in patient education, which is essential for early detection and prevention [36]. Liquid biopsy technologies and other new surveillance techniques are being investigated, highlighting the significance of continuous assessment and international partnerships for improved care and research prospects in this high-risk group [74].

Approximately two-thirds of patients who are suffering from hereditary cancer syndromes like CS express severe difficulty, and they frequently have high levels of worry, sadness, and distress. Female patients with cancer are especially vulnerable to depression. Among them, anxiety and depression are linked to factors like age, unemployment, and previous treatment. It is advised that distress screening be included in national guidelines. Raising awareness of the psychological challenges experienced by patients and their families is essential as genetic diagnostics develop [75]. 

## 4. Conclusions

CS is a disorder caused by mutations in the PTEN gene, which functions as a tumour suppressor gene. Mutations in this gene disrupt the normal regulation of cell growth and division, generating a wide array of clinical presentations and giving rise to diverse manifestations that span from benign to malignant, affecting nearly every system within the human body, particularly the skin, the mucous membranes, the breast, the thyroid, the gastrointestinal tract, and the central nervous system. Management of CS involves regular surveillance and screening for associated malignancies, with the multidisciplinary team playing a major role. 

Genetic counselling and testing are recommended in case of a suggestive family history to assess the risk and provide appropriate guidance, as early detection and comprehensive management strategies are crucial in improving outcomes and quality of life for patients with CS. 

This syndrome is considered rare, and its precise incidence remains unknown among other important factors, which affects individuals experiencing challenges in obtaining an accurate diagnosis and accessing appropriate medical care. 

We urge the academic community to persist in researching all aspects related to CS/PHTS. The literature is notably inconsistent in reporting the frequencies and occurrences of the disorders, as mentioned above, adding an element of bias and uncertainty when looking back at the available research. With this review, our aim is to highlight the varied presentations seen in CS patients, along with the absence of standardized data regarding CS.

## Figures and Tables

**Table 1 medicina-60-00767-t001:** CS manifestations encountered in the literature.

Affected Organ/System	Disorders Encountered	Affected Organ/System	Disorders Encountered
Breast disorders	- Breast cancer - Fibrocystic breast disease	Haematological and vascular disorders	- Arteriovenous fistulas - Haemangiomas (e.g., cavernous ones) - Vasculitis - Gastric tube-related anaemia - Haemolytic anaemia
Endometrial disorders	- Endometrial cancer - Uterine fibroids	Immune disorders	- Lymphopenia - Aberrant B and T cell homeostasis and function (e.g., thymic lymphoid hyperplasia) - Hypogammaglobulinemia - Dysgammaglobulinemia
Gastrointestinal disorders	- Colon cancer - Colorectal polyposis (different histologies) - Papillomatosis (buccal, lingual, gingival, labial mucosa, and pharynx) - Oesophageal glycogenic acanthosis - Gastric cancer - Colitis - Eosinophilic esophagitis	Neurodevelopment disorders	- Macrocephaly - Developmental delay - Autism spectrum disorders - Intellectual disability - Epilepsy
Dermatological disorders	- Melanoma - Papillomatosis - Oral fibromas - Acral keratosis - Lipomas - Trichilemmomas - Penile freckling - Neuromas - Sclerotic fibromas - Café-au-lait spots - Vitiligo - Acrochordons - Acanthosis nigricans	Respiratory disorders	- Airway obstruction due to thymic lymphoid hyperplasia - Pulmonary cysts - Papillomatosis (larynx)
Thyroid disorders	- Multinodular follicular adenomas - Nodular hyperplasia - Adenomatous nodules - Hashimoto thyroiditis - Lymphocytic thyroiditis - C cell hyperplasia - Goitre - Epithelial thyroid cancer (follicular and papillary types)	Metabolic disorders	- Obesity - Insulin dysregulation - Elevated blood levels of lactate and inflammatory markers
Renal disorders	- Renal cysts - Renal cell carcinoma	Nervous system disorders	- Lhermitte–Duclos disease - Meningiomas

**Table 2 medicina-60-00767-t002:** International Cowden Syndrome Consortium Criteria for PHTS/CS.

Major Criteria	Minor Criteria
Lhermitte–Duclos disease	Genitourinary tumours (RCC) or malformations
Thyroid cancer	Lipomas
Macrocephaly	Fibromas
Breast cancer	Mental retardation
	Fibrocystic disease of the breast
	Gastrointestinal hamartomas
	Other thyroid lesions, such as goitre
	Mucocutaneous lesions or palmoplantar keratosis can meet the criteria for CS alone if six or more lesions are present

**Table 3 medicina-60-00767-t003:** NCCN Criteria for PHTS/CS, version 3.2024.

Major Criteria	Minor Criteria
Breast cancer	Autism disorder
Endometrial cancer (epithelial)	Colon cancer
Thyroid cancer (follicular)	Oesophageal glycogenic acanthoses
Gastrointestinal hamartomas (includes ganglioneuromas but excludes hyperplastic polyps, ≥3)	Lipomas
Lhermitte–Duclos disease (adult)	Intellectual disability (IQ ≤ 75)
Macrocephaly (≥97th percentile)	Renal cell carcinoma
Macular pigmentation of the glans penis	Testicular lipomatosis
Multiple mucocutaneous lesions (any of the following):	Thyroid cancer (papillary or follicular variant of papillary)
- Multiple trichilemmomas (≥3, at least one biopsy-proven) - Acral keratosis (≥3 palmoplantar keratotic pits and/or acral hyperkeratotic papules) - Mucocutaneous neuromas (≥3)	Thyroid structural lesions (e.g., adenoma, multinodular goiter)
- Oral papillomas (particularly on tongue and gingiva), ≥3, or at least one biopsy-proven or dermatologist-diagnosed	Vascular anomalies or malformations (including multiple developmental venous anomalies)

**Table 4 medicina-60-00767-t004:** ERN GENTURIS Cancer Surveillance Guidelines for individuals with PHTS/CS.

Cancer	Surveillance Method	Interval	FROM AGE
Breast	MRI	Annually	30 years
	Mammography	Every 2 years	40 years
Thyroid	Ultrasound	Annually	18 years ^1^
Renal	Ultrasound	Every 2 years	40 years
Colorectal	Baseline colonoscopy	-	35–40 years
Melanoma	Baseline skin examination ^2^	-	30 years
Endometrial ^3^	Not recommended	-	-

^1^ Moderate evidence for the age of commencement of surveillance. ^2^ Consider further surveillance as required. ^3^ Consider surveillance as part of clinical trials.

**Table 5 medicina-60-00767-t005:** NCCN Cancer Surveillance Guidelines Version 3.2024 for individuals with PHTS/CS.

Cancer	Surveillance Method	Interval	FROM AGE
Breast	Self-examination	Monthly	18 years
	Clinical examination	Semiannually	25 years OR 5–10 years earlier than the earliest known breast cancer in the family (whichever comes first)
	MRI and mammography	Annually	30 years OR 10 years earlier than the earliest known breast cancer in the family (whichever comes first)
Thyroid	Ultrasound	Annually	7 years
Renal	Ultrasound	Annually or every 2 years	40 years
Colorectal	Baseline colonoscopy	Every 5 years	35 years OR 5–10 years earlier than the earliest known colorectal cancer in the family (whichever comes first)
Melanoma	Baseline skin examination	Annually	-
Endometrial	Patient education regarding the symptoms and the evaluation of these symptoms should include an endometrial biopsy.	-	35 years
	Endometrial biopsy	Annually or every 2 years	

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
