# Peer review of "Insights into Clinical Disorders in Cowden Syndrome: A Comprehensive Review"

_medicina, 2024, doi:10.3390/medicina60050767_

Round 1

Reviewer 1 Report

Comments and Suggestions for Authors

The authors have submitted a review article of illustrating a current knowledge regarding pathophysiological and clinical aspects of Cowden syndrome, a part of the PTEN Hamartoma Tumour Syndrome. The authors searched a range of eligible literature, from well-known classical, and latest research regarding an association of the phenotypes with possible genotypes, which are primarily attributed to the current observations of the disorder. The authors discussed the beneficial availability of a variety of diagnosis and the physiological properties which develop the states of the disease situation, resulting in reliable perspectives. This issue is of interest, and impact of their review is strong. My overall concern with the review describing the current available data regarding beneficial availability of the clinical aspects listed in this review is that information provided may offer something substantial that helps advance our understanding of the disorder which draws novel treatment procedures available in clinic in the future. The reference list may be useful for readers who are interested in this issue.

Readers are considered to know easily the take-home message from the manuscript, but in the conclusion section, the readers could not find the message because the section was consisted of repeated sentences come from abstract and the introduction sections. To strengthen authors’ perspectives, the authors are strongly recommended to re-written the conclusion sections thoroughly to emphasize pathophysiological and clinical aspects of Cowden syndrome. In addition, an additional table demonstrating the suggested characteristics for Cowden syndrome should be incorporated in the revised version because the readers who are interested in this topic need a take-home message from this review.

Author Response

1. Summary

Thank you very much for taking the time to review this manuscript. Please find the detailed responses below and the corresponding revisions and corrections highlighted in the re-submitted files.

2. Point-by-point response to Comments and Suggestions for Authors

Comments 1: The authors searched a range of eligible literature, from well-known classical, and latest research regarding an association of the phenotypes with possible genotypes, which are primarily attributed to the current observations of the disorder. The authors discussed the beneficial availability of a variety of diagnosis and the physiological properties which develop the states of the disease situation, resulting in reliable perspectives. This issue is of interest, and impact of their review is strong. 

Response 1: Thank you for pointing this out.

Comments 2: My overall concern with the review describing the current available data regarding beneficial availability of the clinical aspects listed in this review is that information provided may offer something substantial that helps advance our understanding of the disorder which draws novel treatment procedures available in clinic in the future.

Response 2: Thank you for pointing this out.

Comments 3: The reference list may be useful for readers who are interested in this issue.

Response 3: Thank you for pointing this out.

Comments 4: Readers are considered to know easily the take-home message from the manuscript, but in the conclusion section, the readers could not find the message because the section was consisted of repeated sentences come from abstract and the introduction sections. To strengthen authors’ perspectives, the authors are strongly recommended to re-written the conclusion sections thoroughly to emphasize pathophysiological and clinical aspects of Cowden syndrome.

Response 4: Thank you for bringing this to our attention. Consequently, we have adjusted the conclusion to underscore this aspect. We have included an overview of the pathophysiological and clinical aspects of Cowden Syndrome and incorporated commentary on genetic counseling and testing. Additionally, we have appended a final remark intended to serve as the synthesis of our article, urging the academic community to persist in researching this syndrome, as our review emphasizes its significance. (See the updated text in the manuscript - Lines 573 - 594)

Comments 5: In addition, an additional table demonstrating the suggested characteristics for Cowden syndrome should be incorporated in the revised version because the readers who are interested in this topic need a take-home message from this review.

Response 5: Thank you for bringing this to our attention. Accordingly, we have integrated a discussions section into the article, serving as the third segment, which comprises the following subsections: 3.1. Overview of CS Clinical Disorders, 3.2. Diagnostic Criteria for PHTS/CS, 3.3. Cancer Surveillance Guidelines for PHTS/CS Patients, and 3.4. Screening and Psychological Implications of CS (formerly section 3 of the article). In subsection 3.1., we have provided a comprehensive review of all manifestations documented in the literature for CS, presented in a structured table format based on affected organs or systems. These manifestations have been condensed into a schematic form after detailed description in section 2. Section 3.2. outlines the diagnostic criteria established by the International Cowden Syndrome Consortium and NCCN, demonstrating to readers that despite the wide array of manifestations reported in the literature, not all are considered diagnostic criteria. Additionally, we have included commentary on the evolution of these criteria over time. In section 3.3., we have presented perspectives on cancer surveillance in CS patients from ERN GENTURIS and NCCN, engaging in discussion and comparison of these guidelines. Furthermore, we have provided insights into manifestations not covered by these surveillance guidelines. (See the updated text in the manuscript - Lines 449 - 548 )

Reviewer 2 Report

Comments and Suggestions for Authors

This is a review in the manifestation of PTEN hamartoma syndrome and goes through all of their manifestations.  It is a useful review.

two sections or tables that would make this review more comprehensive would be a table for:

1. different diagnostic criteria for PTHS (NCCN vs PTEN Consortium and other)

2. Different surveillance recommendations (AACR, NCCN, GENTURIS etc).

Comments on the Quality of English Language

Minor comments on english. 

"youthful, wildtype, carry," use updated cancer genetics lexicon PMID 27657676

Author Response

1. Summary

Thank you very much for taking the time to review this manuscript. Please find the detailed responses below and the corresponding revisions and corrections highlighted in the re-submitted files.

2. Point-by-point response to Comments and Suggestions for Authors

Comments 1: This is a review in the manifestation of PTEN hamartoma syndrome and goes through all of their manifestations. It is a useful review.

Response 1: Thank you for pointing this out.

Comments 2: two sections or tables that would make this review more comprehensive would be a table for:

1. different diagnostic criteria for PTHS (NCCN vs PTEN Consortium and other)

2. Different surveillance recommendations (AACR, NCCN, GENTURIS etc).

Response 2: Thank you for bringing this to our attention. Accordingly, we have integrated a discussions section into the article, serving as the third segment, which comprises the following subsections: 3.1. Overview of CS Clinical Disorders, 3.2. Diagnostic Criteria for PHTS/CS, 3.3. Cancer Surveillance Guidelines for PHTS/CS Patients, and 3.4. Screening and Psychological Implications of CS (formerly section 3 of the article). In subsection 3.1., we have provided a comprehensive review of all manifestations documented in the literature for CS, presented in a structured table format based on affected organs or systems. These manifestations have been condensed into a schematic form after detailed description in section 2. Section 3.2. outlines the diagnostic criteria established by International Cowden Syndrome Consortium and NCCN, demonstrating to readers that despite the wide array of manifestations reported in the literature, not all are considered diagnostic criteria. Additionally, we have included commentary on the evolution of these criteria over time. In section 3.3., we have presented perspectives on cancer surveillance in CS patients from ERN GENTURIS and NCCN, engaging in discussion and comparison of these guidelines. Furthermore, we have provided insights into manifestations not covered by these surveillance guidelines. (See the updated text in the manuscript - Lines 449 - 548)

3. Comments on the Quality of English Language: Minor comments on english. "youthful, wildtype, carry," use updated cancer genetics lexicon PMID 27657676

Response on the Quality of English Language: Thank you for bringing this to our attention. We appreciate your comment, and as a result, we have revised the relevant text to ensure clarity for our readers without any potential for misunderstanding. (See the updated text in the manuscript - Lines: 105, 122, 161-162, 196-197, 565, 567-568, 570)

Round 2

Reviewer 1 Report

Comments and Suggestions for Authors

The authors have addressed properly all the issues raised by reviewers including me. I have no more comments, and now recommend that this manuscript is acceptable for publication in Medicina.